# Consumer Motivation behind the Use of Ecological Charcoal in Cameroon

**Ahmed Moustapha Mfokeu [1,2], Elie Virgile Chrysostome [3], Jean-Pierre Gueyie [4,*] and Olivier Ebenezer Mun Ngapna [5]**

1. Faculty of Economics and Management, University of Yaoundé II, Soa, Yaoundé P.O. Box 1365, Cameroon
2. Cedimes Institute USA, Plattsburgh, NY 12901, USA
3. Ivey Business School, University of Western Ontario, London, ON N6G 0N1, Canada
4. Department of Finance, School of Management, University of Quebec in Montreal, Montreal, QC H3C 3P8, Canada
5. Department of Finance and Accounting, University of Dschang, Dschang P.O. Box 96, Cameroon
* Correspondence: gueyie.jean-pierre@uqam.ca

**Abstract:** Climate change and global warming are amplified by pollution and deforestation. For this reason, governments around the world meet every year to find ways to reduce pollution and deforestation and ensure sustainable development. The use of clean energy, particularly ecological charcoal, appears to be an appropriate solution in developing countries. The main objective of this research is to assess the motivations for the consumption of ecological charcoal in Cameroon, using a quantitative approach based on Partial Least Squares Structural Equation Modeling (PLS-SEM). Data were collected from 525 households in the cities of Yaoundé and Douala, Cameroon. The results show that the desire to protect the environment (ecological sensibility), the desire to reduce the energy costs of cooking (economic sensibility), the need to improve health and security, and the desire to enhance the quality of meals and to preserve the cleanliness of pots are all determinants in the consumers' choice to use ecological charcoal. These results are refreshing. In Cameroon, in addition to its economic value, the massive consumption of ecological charcoal will contribute to a reduction in household waste management problems in cities and municipalities, while preserving the environment.

**Keywords:** consumer motivation; green energy; ecological charcoal; ecological sensitivity; economic sensitivity

## 1. Introduction

In sub-Saharan Africa, 80% of the domestic energy needed for cooking, lighting and water heating is provided by wood that may or may not be transformed into charcoal, and to a lesser extent by other types of biomass such as dung and crop residues (International Energy Agency 2016) [1]. Burning fuelwood produces a large amount of smoke and toxic gases that have a negative impact on health (World Health Organization (WHO) 2021) [2]. It causes respiratory, heart and eye damage (WHO 2021) [2]. According to WHO (2021) [2], the inefficient burning of woody biomass is responsible for acute respiratory infections causing over 600,000 premature deaths per year in Africa (WHO 2021) [2]. However, less dangerous energy solutions for cooking exist. These include improved stoves, solar cookers and ecological charcoal.

Ecological charcoal is made from organic waste (e.g., cassava, potato and macabo peelings; maize cobs; rattan, coconut, sugar cane waste and more generally from food waste) (Idriss et al. 2021) [3], mature trees (e.g., olive trees), and wood residue and waste. It is a way of partially solving problems associated with waste management in cities. The consumption of ecological charcoal may then be seen as an ecological act aimed to protect the environment. A closely related issue is ethical consumption. Ethical consumption

has been studied from several relatively complementary angles: fair trade (De Ferran and Grunert 2007) [4], boycotts (Smith 1990) [5], socially responsible consumption (François-Lecompte and Valette-Florence 2004; Ozcaglar 2005) [6,7] and ecological consumption (Gierl and Stumpp 1999) [8]. These studies [4–8] have a specific context and do not focus on the consumption of products made from waste.

The consumption of ecological charcoal is expanding within the Cameroonian population. This leads us to question the forces behind such a move. Thus, the main objective of this research is to assess the motivations of consumers to use ecological charcoal, which materials have been listed above. More specifically, the underlying research questions are: Does ecological sensitivity influence the consumption of ecological charcoal? Does economic sensitivity influence the consumption of ecological charcoal? What are the other socio-economic factors that favor the use of ecological charcoal in Cameroon? The Cameroonion context needs to be analyzed because of its relevance in curbing deforestation and climate change (Bas et al., 2022) [9].

Analyses are conducted using Partial Least Square Structural Equation Modeling (PLS-SEM). Data are collected by a questionnaire, from 525 households in the cities of Yaoundé and Douala in Cameroon. The results show that the desire to protect the environment (ecological sensibility), the desire to reduce the energy costs of cooking (economic sensibility), the need to improve health and security, and the desire to enhance the quality of meals and preserve the cleanliness of pots are determinants in the consumer's choice to use ecological charcoal (Basha et al., 2015; Weerasiri and Koththagoda, 2017; Tize et al., 2020; Kodji et al., 2021) [10–13].

The paper continues as follows: Section 2 presents issues regarding the ecological charcoal use in Cameroon, while Section 3 presents the conceptual framework and hypotheses development. Section 4 provides details about the data collection and methodology. Section 5 reports and discusses the results, and Section 6 concludes the paper.

## 2. Issues of Ecological Charcoal in Cameroon

According to the World Bank (2019) [14], Cameroon is a lower-middle-income country with a population of over 25 million of inhabitants (2018). The country is presented as Africa in miniature. Like most African countries, Cameroon is increasingly facing important environmental issues, one of which is pollution. The major forms/types of pollution include indoor and outdoor air pollution, land pollution and water contamination. Water contamination is a major challenge. Its main causes are linked to the poor management of household waste in the cities due to uncontrolled urbanization. The use of streams as waste dumps or as emptying places for household waste is a current practice that leads to groundwater pollution in the main cities, such as Yaoundé and Douala, and to waterborne diseases. For example, in September 2010, more than 385 Cameroonians lost their lives due to cholera (The cause of soil and water contamination is not limited to household waste. Industrialization has led to the disposal of heavy metals in the environment, such as zinc (II), a pollutant which a high toxicity to plant animal and human life (Fonseca et al., 2006 [15]). Many technologies have been developed to remove metal ions from water, but adsorption is the most effective way. Adsorbents can be produced from agricultural waste. Among them, Ravindran et al. (2018) [16] showed that the absorption of zinc (II) is feasible using palm shell-based activated carbon. The adsorbent obtained from palm shell-based activated carbon is more efficient compared to other types of adsorbent produced from agricultural waste).

About 2.8 billion people (38% of the world population and nearly 50% of the population of developing countries) depend on biomass fuels for their cooking and heating needs (IEA 2016) [1]. These fuels are mainly firewood, charcoal, manure and crop residues. According to the Food and Agriculture Organization (FAO) of the United Nations (2017) [17], biomass provides more than 70% of the primary energy consumed by African households exclusively for cooking, heating and lighting. This percentage is even higher in Cameroon, which is one of the largest forest countries in the Congo Basin.

The importance of Cameroon's forests to the biosphere is widely unknown, but it is comparable to the Amazon and the boreal forests. Bele et al. (2011) [18] reported that Cameroon's forests cover approximately 23 million hectares, representing almost 50% of the country's total land area. Furthermore, the authors estimate that Cameroon is home to some 8260 plant species (of which 156 are endemic) and approximately 2000 wildlife species, making it the fifth largest country in Africa in terms of biodiversity. Thus, preserving Cameroon's forests is an obligation and a duty for humanity (FAO and UNEP 2020) [19].

Yet Daurella and Foster (2009) [20] show that 82.6% of Cameroonian households use fuelwood as their first source of energy, whether in the form of charcoal, firewood, sawdust or woodchips. The wood energy sector is the first source of domestic energy accessible to the population, especially those living in rural areas. Indeed, residents often burn ligneous material or convert it into ordinary charcoal by carbonization. This conversion sometimes leaves considerable environmental damage. In addition, deforestation causes flooding in the northern region, and crop loss and the disruption of agricultural activity schedules in the south (See FAO, 2021 [21], for examples of disasters and crises on agriculture). This is a factor aggravating the problems of famine (World Bank 2015) [22].

On the other hand, the daily production of waste in a country's urban areas continues at a bewildering rate with a rather insufficient absorption capacity. A city such as Douala produces 3500 tons of waste per day. Yet, just 54% of this waste is collected (Bosangi 2020) [23]. Hence, on one hand, the objective of the public authorities is to provide an alternative energy solution to charcoal and wood at a lower cost for the most disadvantaged populations. On the other hand, it is to recover the waste, which abounds in the major cities of Cameroon. This was certainly the reason for the development of the National Waste Management Strategy (2007–2015) (Ngambi 2015) [24].

An immediate substitute for firewood and charcoal could be ecological charcoal, which is made from agricultural residues, household waste, dry biomass from grasses and a binder (starch flour or clay). Ecological charcoal, as its name suggests, has an overall positive impact on the environment. It makes it possible to limit deforestation (due to the use of non-timber plant matter and agricultural residues) (Tize et al. (2020)) [12]. It enhances the depollution and sanitation of urban environments (due to the use of biodegradable waste) (Ravindran et al., 2018) [16] and it represents a supply of domestic energy for cooking that is less time-consuming and more clean. The only drawback to ecological charcoal is that it produces much more ash compared to wood. On the other hand, though, this ash is an excellent fertilizer.

The production of ecological charcoal is at the forefront of the circular economy. Ecological charcoal costs a little less than conventional charcoal. Additionally, it takes a longer time to burn as compared to conventional charcoal.

Currently, ecological charcoal is not well known among Cameroonians. Its consumption is marginal, due to its rarity within the country and low promotion. However, its sales are on the rise in the cities of Douala and Yaoundé. It is distributed through individuals (retailers), mini supermarkets such as "Mbom" in Akwa and Douala, and supermarkets such as Carrefour (Douala and Yaoundé) and U in Douala.

The production remains limited. German Technical Cooperation (GIZ) has been working since 2016 to develop a more sustainable wood energy sector in the East. The project supports charcoal groups and the Network of Charbonniers des Concessions Forestières de l'Est (RECHACOFEST) to produce "greener charcoal". In 2017, the regional agency of the National Employment Fund held a seminar in Ebolowa in the southern region about the production of ecological charcoal, with around fifty charcoal makers from the city. Similarly, residents of the locality of Minawao in the far north region have benefited from training in the manufacture of ecological briquettes, with funding from the High Commission for Refugees (HCR), in partnership with the Lutheran World Federation (LWF). On the site of Cinq-Clous (a district) in Ebolowa, ecological charcoal is now selling at affordable local prices. At retail, a 10 L bucket is sold for CFA 500 (USD 0.79 on 22 December 2022), while a 15 L bucket is sold for CFA 1000 (USD 1.58). At wholesale prices, the cost of a 50 kg

bag fluctuates between CFA 3000 and 5000 in the south. In the far north of the country, this same bag is sold for just over CFA 10,000 (USD 15.8) (we thank the Ministry of environment, protection of nature and sustainable development of Cameroon for providing this useful information). Biochar Kmer cooperative and Kermit Ecology are among the known producers.

The choice of ecological charcoal as source of energy for cooking food and as a solution for environmental protection is not new in Cameroon. It has been the subject of several studies, particularly related to the far north. According to Kodji et al. (2021) [13], the total demand for wood fuel for this region is evaluated at 1.38 million tons, the equivalent of 23,400 terajoules (TJ), whereas it produces 800,000 tons of sustainably exploitable wood fuel. It thus faces a deficit of 580,000 tons of wood fuel that exposes the region to a true ecological disaster. Kodji et al. (2021) [13] concludes that: "A substitutional energy at low cost and accessible would be ecological charcoal. The Region has an estimated annual production capacity at 2.65 million tons in average of ecological charcoal, the equivalent of about 60,000 TJ. Such production could cover the annual needs of 10 million inhabitants right up to 2050 in the Region." Tize et al. (2020) [12] also evaluated the production capacity of ecological charcoal, but only for the city of Maroua in the far north region of Cameroon. The authors found that "the annual potential of white coal production in Maroua indicates that 839.54 tons can be obtained at the National Advanced School of Engineering and the Waste Treatment Centre of HYSACAM. This potential could cover the annual cooking energy needs of 3872 people in the city of Maroua, and preserve 196 hectares of firewood collection area".

## 3. Conceptual Framework and Hypotheses Development

"Consumer behaviour is the study of all the actions of the individual that are directly related to the purchase and use of economic goods and services, including the decision-making process that precedes and determines these actions" (Kotler 1998) [25]. Thus, the study of consumer behavior is concerned with the feelings, actions, reasons, motivations, deeds and gestures of individuals. This study of consumer behavior analyzes the decision-making processes that lead individuals to spend their resources (Van Vracen and Janssens-Umflat 1994) [26]. The work that deals with the ecological behavior of consumers can be divided into two groups: those with an individual approach to explaining consumption behavior, and those with a contextual approach to explaining the ecological consumption of individuals. With regard to the work on energy transition, Maslow's hierarchy of needs (1954) can explain consumer choices [27].

### 3.1. The Individualistic Approach to Green Consumption

Previous work on consumer ecological consumption behavior reports that several personal variables intervene in the individual's commitment to carry out pro-environmental actions (Giannelloni, 1998; Teixeira et al., 2022) [28,29]. For example, the consumer's environmental concern, knowledge and beliefs about the environment have a direct and indirect influence on the person's intention to behave in a pro-environmental way (Pagiaslis and Krontalis 2014) [30]. Consumer knowledge can be objective or subjective. It represents what consumers know about the environment and about environmental problems, and the actions that can be taken to counter these problems. Beliefs can be general (about the environment and ecological consumption) or specific (about a particular product or action). Beliefs may also relate to the perceived effectiveness of pro-environmental actions (D'Astous and Legendre 2009) [31]. Zaiem (2005) [32] argues that there is a positive link between environmental knowledge, ecological sensitivity and the ecological behavior of the consumer. Binninger and Robert (2008) [33] maintain that there is a strong association between environmental issues and sustainable development in the minds of consumers, which shows that these customers are ready for the advent of economic policies strongly anchored in sustainable development.

According to Klöckner (2013) [34], perceived efficacy has a direct influence on the behavior of individuals. In other words, individuals will be more likely to engage in environmentally friendly behavior if they feel they have the ability or power to make a difference through their actions. Furthermore, the purchase of environmentally friendly products by consumers can be influenced by many factors, including environmental and social awareness or concerns, socio-demographic characteristics, political ideologies, collective values, brand preference, affinity for nature, cost–benefit analyses, trust in advertisements and perceived values and risks (Durif et al., 2011; Bougherara and Combris 2009) [35,36]. Self-expression can be understood as the ability of a product or service to project the image of oneself (to others but also to oneself) and to play a role in social communication as a reflection of one's personality (Aurier et al., 2004) [37]. Self-expression can influence the value of a good for consumers, and therefore its consumption. The consumption of environmentally friendly products can thus contribute to projecting a certain self-image.

### 3.2. The Contextual Approach to Green Consumption

With regard to the contextual approach, several contextual factors can facilitate or constrain the adoption of an environmental behavior. The constraints can be so strong and costly in terms of time, money and effort that it would be difficult for the consumer to refuse to adopt ecological behavior, despite the person's personal motivations (Giannelloni 1998; Corraliza and Berenbguer 2000) [28,38]. Thus, the availability of green products, infrastructure, technical facilities, regulations and product characteristics can directly or indirectly influence behavior change (Steg and Vleg 2009) [39].

For Lalonde (2015) [40], the greening of the economy and the creation of a global dimension of public affairs are indispensable measures to be taken by the member countries of the UN to resolve the climate crisis. The greening of the economy refers to the integration of its flows into natural cycles, and the recognition of the value of natural capital. Policymakers are beginning to recognize that nature is a factor of production no less decisive than capital and labor. The green economy spares natural resources. In the past, natural resources were plentiful while labor was scarce, but today, the opposite is true. This justifies the shift in taxation towards penalizing resource extraction and removing the burden on labor. The green economy reduces extractive activities, avoids waste, builds on natural processes, seeks efficiency and strives for a circular economy. Zaiem (2005) [32], has shown that the ecological behavior of consumers varies according to the socio-demographic and psychographic variables present in a country.

Based on the theory of planned behavior (Ajzen 1991) [41], several studies have shown that individuals make reasoned choices and opt for solutions that bring them the most advantages and the least disadvantages. Even when individuals are aware of the importance of ecological behavior, they are not always prepared to accept the negative consequences, such as a loss of time or the consumption of expensive or lower-quality products (Carrigan and Attala 2001) [42]. This theory has been shown to be effective in explaining various types of ecological behavior, such as choice of travel mode, recycling, composting of waste, meat consumption or pro-environmental behavior in general (Steg and Vlek 2009) [39].

Still on the topic of ecologically friendly products, the determinants of the purchase intention of organic food is a topic that has been widely investigated in the literature [9,10,29,43–49]. Several of these determinants fall within the individualistic and contextual approaches presented above.

### 3.3. The Energy Transition, Fuel Choice and Maslow's Hierarchy of Needs

Maslow's hierarchy of needs (Maslow, 1954) [28] is a hierarchical representation of human needs, which shows that one can only reach the top of the hierarchy if their basic needs are satisfied. The motivation for choosing an energy solution corresponds to the motivation for satisfying a need that lies on the energy ladder model illustrated in Figure 1. In the mid-1980s, the field of economics related to the energy transition in developing

countries was structured around this energy ladder model. It was formalized by Hosier and Dowd (1987) [50], based on a correlation observed at a macro level between electricity consumption and the GDP (Robert, 2021) [51].

To theorize household energy consumption behavior, Reddy (2000) [52] and Van der Kroon et al. (2013) [53] use standard consumption theory, considering that individuals or households make a consumption choice of one type of energy source from an available mix. This choice maximizes their income-constrained utility according to the "objective" physical characteristics of different energy sources (cleanliness, individual exposure to pollutants, ease of use and energy efficiency) (Reddy 2000; Clancy 2006; Hiemstra-Van der Horst et al. 2008) [52,54,55].

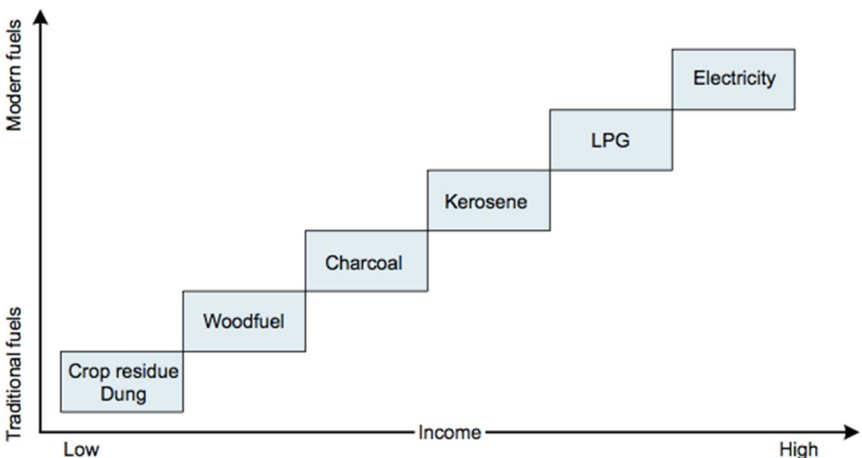

**Figure 1.** Schematic representation of the energy scale (Smith et al. 1994) [56].

The result is a hierarchical laddering of these sources, with each income level corresponding to a dominant energy type that, while not the only one available, maximizes household utility (Hosier and Dowd 1987; Reddy and Reddy, 1994) [50,57]. In the case of low incomes, households stick to "traditional" energy sources that induce a large number of usage constraints. When incomes rise, households make a fuel-switching transition to a higher level (Smith et al., 1994) [56]. The switching process ends when households reach an income level that allows them to use "modern" forms of energy, such as liquefied petroleum gas and electricity (Leach and Mearns 1988; Leach 1992; Clancy 2006; Mekonnen and Kohlin, 2009) [54,58–60]. These are more efficient and useful, but also more expensive to purchase (Robert, 2021) [51].

Until the 2000s, economic studies focused on the empirical verification of the energy ladder (Barnes and Qian 1992; Leach 1992; Smith et al., 1994; Reddy 1995) [56,59,61,62]. These authors provided evidence for this model and proposed new variants of it: a "rural energy ladder" (Barnes and Qian 1992) [61] and a "ladder of energy demand" (Foley 1995) [63]. Finally, they led to its reappraisal by highlighting a mode of consumption using "multiple fuels" (Davis, 1998) [64]. Despite an increase in income, researchers found that households may decide to continue to consume "traditional" energy sources (Leach 1992) [59] or adopt an energy mix composed of different sources (Masera and Navia 1997; Masera et al., 2000; Roy 2000; Thom 2000; Mestl et al., 2009) [65–69]. Some studies use the concepts of "multiple fuel use" and "fuel stacking". This concept thus occupies a prominent place in the literature (Kowsari and Zerriffi 2011) [70].

In parallel to the energy transition, several studies have focused on analyzing the factors affecting household energy choices. These studies have highlighted the influence of income (Fitzgerald et al. 1990; Elias and Victor 2005; Narasimha Rao and Reddy 2007; Kwakwa et al., 2013; Choumert-Nkolo et al., 2019) [71–75], household size and composition (Dewees 1989; Heltberg et al. 2000; Heltberg 2005; Reddy 2007; Pandey and Chaubal 2011; Nepal et al., 2011) [76–81], the level of education of household members and the gender of household heads (Israël 2002; Heltberg 2004, 2005; Pachauri and Jiang 2008; Pandey and Chaubal 2011) [80,82–84], as well as cultural preferences (Fitzgerald et al., 1990; Masera et al., 2000; Heltberg 2004; Xiaohua and Zhenmin, 2003; Guptaa and Kohlin, 2006; Hou et al., 2017) [66,71,83,85–87].

### 3.4. Hypotheses Development

A review of the literature on the motivations for the consumption of ecological charcoal allows us to identify the main motivations for the use of ecological charcoal.

### 3.5. Environmental Awareness and the Use of Ecological Charcoal

To explain ecological consumption behavior, the concept of values is essential to demonstrate the structures and changes that are at the heart of societies and individuals (Durkheim 1893, 1897; Weber 1905) [88–90]. Indeed, values are the basis of what explains motivations in humans as well as in different communities (Schwartz 2006) [91]. As Manaktola and Jauhari (2007) [92] suggest, consumers will prefer a product that is related to their personal values.

Environmental concern is generally related to rational behavior that preserves ecosystems. Most measures of environmental concern incorporate items relating to all three components of attitude (cognitive, affective and conative). The cognitive component reflects subjective knowledge about environmental problems and the actions taken to improve them. The affective component reflects emotional responses to perceived environmental problems, and the conative component is explained by a tendency to make a personal contribution to improving the environment (Dembkowski and Hammer-Lloyd 1994) [93].

It is evident from the work of Maloney and Ward (1973), Grunert, (1993), Roberts (1996), Li (1997), and Chan and Lau (2000) [94–98] that there is a positive link between ecological sensitivity and ecological consumption. From the above, the following hypothesis can be formulated:

**Hypothesis 1 (H1).** *Ecological sensitivity positively influences the use of ecological charcoal.*

### 3.6. Household Costs Reduction and the Use of Ecological Charcoal

Several studies based on the theory of planned behavior (Ajzen 1991) [41] have shown that individuals make reasoned choices and opt for solutions that bring them the most advantages and the least disadvantages. Even when individuals are aware of the importance of their ecological behavior, most individuals are not prepared to accept the negative consequences, such as loss of time or the consumption of expensive or lower-quality products (Carrigan and Attala 2001) [42]. This theory of planned behavior has been shown to be effective in explaining various types of ecological behavior, such as the choice of travel mode, recycling, composting of waste, meat consumption or pro-environmental behavior in general (Steg and Vlek 2009) [39]. Several studies have focused on analyzing the factors affecting household energy choices, and have highlighted the influence of income. Income is the main determinant of energy use (Hosier and Dowd 1987; Robert 2021; Reddy 2000; Van der Kroon et al. 2013; Heltberg 2004) [50–53,83]. In these contexts, the energy consumed is mainly collected; it does not incur any monetary expenditure. Where it exists, access to commercial energy markets is difficult. Thus, the following hypothesis is formulated:

**Hypothesis 2 (H2).** *Economic sensitivity positively influences the use of ecological charcoal.*

*3.7. Health, Security (Reduction in the Risk of Fire) and the Use of Ecological Charcoal*

Much research has identified household air pollution as a major risk to human health, particularly in developing countries where the use of traditional and inefficient fuels is widespread (Duflo and Greenstone 2008) [99]. The negative impacts of exposure to household air pollution resulting from the inefficient use of solid fuels represent the greatest energy-related health risk (Smith et al., 2014) [100]. In India, for example, more than one million premature deaths each year are attributable to household air pollution (Balakrishnan et al., 2019) [101]. According to WHO (2021) [2], the inefficient burning of woody biomass is responsible for acute respiratory infections causing over 600,000 premature deaths per year in Africa. Globally, estimates place the number of deaths caused by household air pollution at between 2 and 4 million per year (Lin et al., 2013; Stanaway et al., 2018) [102,103]. McCracken and Smith (1998) [104] found that Guatemalan households exposed to carbon monoxide from firewood had more health problems than less-exposed households. In a study of Indian households, Duflo and Greenstone (2008) [99] found a significant correlation between symptoms of respiratory illness and the use of traditional stoves. Health problems related to exposure to indoor air pollution can lead to difficulties in carrying out activities such as education, domestic tasks and labour market participation, which in turn generate economic losses, including lost wages and increased medical costs (Bakehe 2021) [105]. Ecological charcoal significantly reduces this air pollution. In addition, it reduces the risks of burns and fire (which means increased security). From the above, the third hypothesis is formulated as follows:

**Hypothesis 3 (H3).** *The desire to improve health and security positively influences a person's use of ecological charcoal.*

*3.8. Preferences Regarding Quality of Meals and Cleanliness of Pots, and Ecological Charcoal*

Concerning food preparation, the diversity of energy situations and choice depends mainly on non-economic factors (Takama et al., 2012) [106], such as meal frequency, consumption patterns, food taste preferences (Leach 1988; Fitzgerald et al., 1990; Heltberg 2005) [58,71,78], ethnicity (Heltberg 2005) [78], and local traditions and institutions (Hiemstra-Van der Horst and Hovorka 2008) [55]. Takama et al. (2012) [106] demonstrated the interdependence of different factors. Some energy transition processes are described as "two-stage processes" (Masera et al., 2000) [66]. Some households prefer wood fires for cooking certain dishes when other energy sources are available. In Africa, most meals are cooked over a wood fire, and that is healthier. Some people think that changing the energy source for cooking does not give the food the same taste. These people demand that some meals be cooked in the traditional way. Taken in this way, the taste preference for food has a negative effect on the use of ecological charcoal. However, is it possible that ecological charcoal results in better quality meals (that can be kept longer), or in meals with better nutritional value? Consumers' perceptions will help answering this question. In addition to quality, an uncontested advantage of ecological charcoal is that it does not blacken the bottom of pots. From this argumentation, the following fourth hypothesis is formulated:

**Hypothesis 4 (H4).** *Preferences for the quality of meals and the cleanliness of pots have a positive effect on the use of ecological charcoal.*

The conceptual model is summarized in Figure 2.

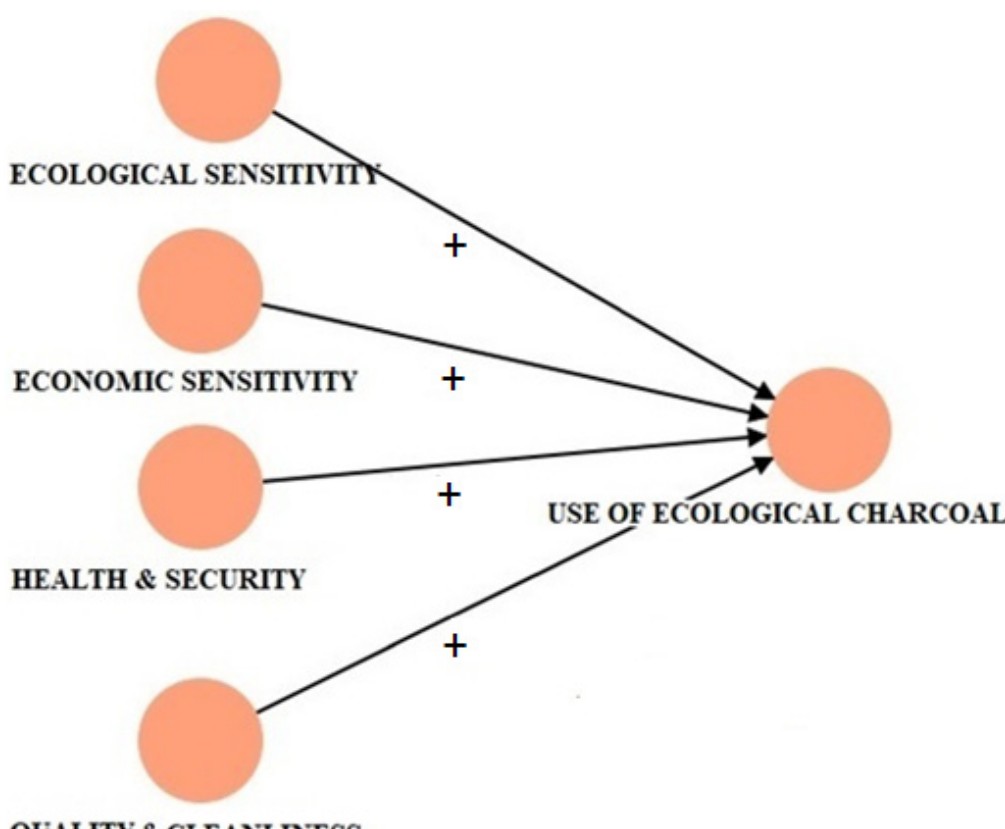

**Figure 2.** Conceptual model.

## 4. Data and Methodology

### 4.1. Sampling and Data Collection

The cities of Yaoundé and Douala were selected for this study. These two cities are the most populated in Cameroon. Yaoundé and Douala are home to populations from all parts of the country, the majority of whom are young. A non-probabilistic sampling method (and more specifically, convenience sampling) is chosen given the lack of an exhaustive directory of people who use charcoal. The study of the consumption behavior of young people with regard to energy sources for cooking seemed to be more appropriate, given that they are probably the segment of the population that should be targeted to modify future consumption behavior, with the aim of promoting sustainable consumption and respect for the environment.

Our data were collected using a closed-ended questionnaire. A 5-point Likert scale ranging from "completely disagree" to "completely agree" serves as the measurement scale for each item. The questionnaire was administered face-to-face and at home or in the workplace to a convenience sample of 525 households in Yaoundé and Douala. Initially, 753 people were interviewed and only 525 questionnaires were deemed usable. The data collection took place in September and October 2022. Following Sarmah et al. (2013) [107], we use a Cochran (1977) [108] formula to estimate our minimum sample size. We started with the case of an infinite population. In this case,

$$no = \frac{Z^2 p(1-p)}{e^2},$$

where $n_0$ is the sample size, $z$ is the selected critical value of the desired confidence level (1.96 for the 95% confidence level), $p$ is the estimated proportion of an attribute that is present in the population, $q = 1 - p$, and $e$ is the desired level of precision or error margin (5%, for the 95% confidence level). In 2023, the population in Cameroon has been estimated at 26.55 million inhabitants. With an average of 4 people per household, this gives us a

total number of 6,637,500 households. Let us assume that about half of this population is concerned by ecological charcoal. Thus $p = 0.5$, and $1 - p = 0.5$. At a 95% confidence level ($z = 1.96$ and $e = 0.05$),

$$no = \frac{1.96^2 * 0.5 * 0.5}{0.05^2} = 384.16.$$

The second step was to adjust this number for the case of a finite sample. Sarmah et al. (2013) [107] used the following formula:

$$n = \frac{n_0}{1 + ((n_0 - 1)/N)},$$

where $n$ is the final minimum sample size, $n_0$ is the sample size derived from the previous equation and $N$ is the population size. This leads us to:

$$n = \frac{384.16}{1 + ((384.16 - 1)/6,637,500)} = 384.13.$$

A large sample size is better than a smaller one. So, we decided to double this minimum sample size at the beginning of our survey. We were finally able to interview 753 people, of which only 525 questionnaires were deemed usable.

The main difficulty encountered was the reluctance of some people to answer the questions. We had to reassure them that the study was purely scientific.

### 4.2. Variables Description

The dependent variable in this research is the use of ecological charcoal. The independent variables are ecological sensitivity, economic sensitivity, health and security, and quality and cleanliness. All these variables are constructs. Their definitions are provided in Table 1, with a description of their indicators (items). Generally, these constructs and items were inspired by the literature.

**Table 1.** Constructs and items descriptions.

| Constructs | Items |
|---|---|
| The use of ecological charcoal (ECOL_US) is defined as the extent to which an individual has used and is ready to use ecological charcoal. | ECOL_US1: I have already used ecological charcoal, and I was satisfied with its use.<br>ECOL_US2: I have already used ecological charcoal, and it meets the expectations that I had before using it.<br>ECOL_US3: I have already used ecological charcoal, and I will use it again.<br>ECOL_US4: I have already used ecological charcoal, and I will recommend it to others. |
| Ecological sensitivity (ECOLO_S) is defined as the extent to which an individual cares about protecting the environment and related issues. | ECOLO_S1: I always put my waste in the bins provided for this purpose/I never it on the ground or pour it into the gutters.<br>ECOLO_S2: I am willing to buy products and/or services that have a positive impact on nature and the environment.<br>ECOLO_S3: I am willing to participate in taking actions that protect nature and the environment.<br>ECOLO_S4: I would be willing to sort waste if waste separation bins were installed near my home.<br>ECOLO_S5: In general, I have a great sensitivity for environmental issues.<br>ECOLO_S6: The use of ecological coal can contribute to making our cities cleaner (better waste management).<br>ECOLO_S7: I would support the establishment of a policy that popularizes and promotes the use of ecological charcoal. |

**Table 1.** *Cont.*

| Constructs | Items |
|---|---|
| Economic sensitivity (ECONO_S) is defined as the extent to which an individual cares about cost savings, and choose to use a low-cost source of energy for cooking. | ECONO_S1: Charcoal burns longer than wood or charcoal. ECONO_S2: My use of ecological charcoal is mainly motivated by the potential cost reduction that this energy source provides. ECONO_S3: My use of ecological coal effectively reduces my energy costs of cooking. ECONO_S4: When I have the choice between several sources of energy, I always choose, the least expensive (I am not ready to pay more to protect nature or the environment). ECONO_S5: My financial conditions force me to always choose the least expensive source of energy. ECONO_SE6: Ecological charcoal promotes job creation in society. |
| Health and security (HEAL_SE) is defined as the extent to which an individual cares about the protection of the environment and related issues. | HEAL_SE1: The use of ecological charcoal has significant beneficial effects on the health of the user/of those around. HEAL_SE2: The use of ecological charcoal reduces the risk of burns for the user/their relatives. HEAL_SE3: The use of ecological charcoal reduces the risk of mortality caused by poisoning by wood smoke or charcoal. HEAL_SE4: It is healthier to use ecological charcoal inside a home than wood or charcoal. HEAL_SE5: In general, the use of clean coal improves the overall health of society. |
| Quality of meals and cleanliness (QUAL_CL) is defined as the desire to keep or improve the quality of meals, as well as to maintain the cleanliness of cooking equipment. | QUAL_CL1: Ecological charcoal does not blacken the bottom of pots. QUAL_CL2: Using ecological charcoal produces better quality meals (example: which can be kept longer). QUAL_CL3: Using ecological charcoal produces meals with better nutritional value. QUAL_CL4: In general, the use of ecological charcoal gives the family greater pleasure in eating cooked food. |

*4.3. Methodology*

The estimates were carried out using Partial Least Square Structural Equation Modeling (PLS-SEM) (Ringle et al., 2015) [109]. This technique is appropriate in exploratory and confirmatory research (Chin 2010; Petter, Straub, and Rai 2007) [110,111]. The development follows two steps: (1) assessment of the measurement model and (2) evaluation of the structural model (Dijkstra and Henseler 2015; Henseler, Ringle, and Sarstedt 2015; Peng and Lai 2012) [112–114].

*4.4. Assessment of the Measurement Model*

In the study, reflective variables per construct are used, which implies that unobserved constructs give rise to the observed indicators. The direction of influence goes from the construct to the indicators, and the indicators observed constitute a reflection or manifestation of the construct. Reliability, convergent validity and discriminant validity are evaluated following the criteria of Fornell–Larcker (1981) [115].

Reliability is evaluated through the construct's Cronbach alpha, Rho_A and composite reliability coefficients. A value above 0.7 is recommended. Convergent validity is evaluated using the Average Variance Extracted (AVE) criterion (Fornell and Larcker 1981) [115]. Sufficient convergent validity involves an AVE of at least 0.5. The discriminant validity is checked through the Fornell–Larcker criterion. It suggests that the square root of the AVEs, along with the diagonal, must be greater than all the values below and left of the AVEs located on the diagonal (Fornell and Larcker, 1981; Hair et al., 2014) [115,116].

*4.5. Assessment of the Structural Model*

The second part of the PLS-SEM methodology is a structural or inner model that specifies the relationships among the constructs (i.e., relating some endogenous latent variables (LVs) to other LVs). The SMART PLS 4 software is used to perform this evaluation.

## 5. Results and Discussion

### 5.1. Descriptive Statistics

Table 2 presents the demographic profile of the respondents.

**Table 2.** Demographic profile of the respondents.

| Profile | Description | Frequency | Percentage |
|---|---|---|---|
| Gender | Male | 118 | 22.47% |
| | Female | 407 | 77.53% |
| Age | 18–34 | 369 | 70.28% |
| | 35 and more | 145 | 29.71% |
| Household size | 1–4 | 145 | 27.61% |
| | 5 and more | 380 | 72.39% |
| Level of education | Unschooled | 18 | 3.42% |
| | Primary school | 23 | 4.38% |
| | Secondary school | 43 | 8.19% |
| | Undergraduate | 182 | 34.66% |
| | Graduate | 259 | 49.33% |
| Employment | Unemployed | 65 | 12.38% |
| | Student | 102 | 19.42% |
| | Maneuver | 148 | 28.19% |
| | Middle management | 96 | 18.28% |
| | Senior management | 113 | 21.53% |
| Monthly revenues | Less than CFA 50,000 | 194 | 36.95% |
| | 50,000–100,000 | 205 | 39. 04% |
| | 100,005–300,000 | 100 | 19.04% |
| | 300,005–500,000 | 22 | 4.19% |
| | 500,005–1,000,000 | 2 | 0.38% |
| | 1,000,005 and more | 2 | 0.38% |

From Table 2, it can be seen that 77.53% of the individuals interviewed are women and 22.47% are men. The high representation of women in the sample is explained by the fact that in the Cameroonian context, women are the most concerned with cooking. Thus, they are the most likely to give an objective opinion on the optimal choice of thermal energy sources to use. The age structure of the sample is close to that of the target population of the study, with 70.28% of the respondents aged 18–34 years. Additionally, 27.61% of the families included in the sample have more than five persons. Students and the unemployed represent 31.81% of the sample, and the participants are on average educated, with 34.66% and 49.33% holding undergraduate and graduate degrees, respectively. The respondents who have very small monthly revenues (less than 50,000 FCFA) represent more than 36.95% of the sample.

### 5.2. Assessment of the Measurement Model

The first step in the PLS-SEM approach is the evaluation of the measurement model that specifies the relationship between the unobserved variable (construct) and the observed items (indicators). Table 3 presents the results of the measurements of the reliability and validity of the items and constructs.

**Table 3.** Construct reliability and validity.

| Construct | Items | Loading | Cronbach's Alpha | Rho_A | Composite Reliability | Average Variance Extracted (AVE) |
|---|---|---|---|---|---|---|
| Use of Ecological Charcoal | ECOL_US1 | 0.944 | 0.957 | 0.960 | 0.969 | 0.887 |
| | ECOL_US2 | 0.945 | | | | |
| | ECOL_US3 | 0.946 | | | | |
| | ECOL_US4 | 0.932 | | | | |
| Ecological Sensitivity | ECOLO_S2 | 0.705 | 0.871 | 0.906 | 0.903 | 0.610 |
| | ECOLO_S3 | 0.797 | | | | |
| | ECOLO_S4 | 0.654 | | | | |
| | ECOLO_S5 | 0.826 | | | | |
| | ECOLO_S6 | 0.819 | | | | |
| | ECOLO_S7 | 0.865 | | | | |
| Economic Sensitivity | ECONO_S2 | 0.847 | 0.760 | 0.764 | 0.820 | 0.609 |
| | ECONO_S3 | 0.862 | | | | |
| | ECONO_S6 | 0.606 | | | | |
| Health and Security | HEAL_SE1 | 0.850 | 0.846 | 0.858 | 0.906 | 0.763 |
| | HEAL_SE3 | 0.853 | | | | |
| | HEAL_SE4 | 0.849 | | | | |
| | HEAL_SE5 | 0.876 | | | | |
| Quality and Cleanliness | QUAL_CL1 | 0.774 | 0.866 | 0.917 | 0.898 | 0.638 |
| | QUAL_CL2 | 0.843 | | | | |
| | QUAL_CL3 | 0.795 | | | | |
| | QUAL_CL4 | 0.755 | | | | |

(ECOLO_S1 = 0.523; ECONO_S1 = 0.594, ECONO_S4 = 0.520, ECONO_S5 = 0.513; HEAL_SE2 = 0.580).

The reliability of each item, assessed through the outer loading, shows that almost all the items retained here have a loading value > 0.7 (Hair et al., 2016) [117]. The item ECOLO_S1 has been removed from the perceived ecological sensitivity because it has a loading lower than 0.6. Three items (ECONO_S1, ECONO_S4 and ECONO_S5) have been removed from the perceived economic sensitivity, because their loading was lower than 0.6. Similarly, one item (HEAL_SE2) has been removed from the construct of health and security.

The items ECOLO_S4 and ECONO_S6 have a loading value > 0.6 but < 0.7. We decided to keep them in the evaluation.

The reliability of the constructs is tested using Cronbach's alpha and the composite reliability. The values of Cronbach's alpha are above 0.7 for all constructs. The values for the composite reliability are all greater than 0.7, as recommended by Henseler, Ringle and Sincovics (2009) [118]. Thus, all our constructs are reliable, according to conventional standards (Fornell and Larcker 1981; Henseler, Ringle and Sincovics 2009; Chin 2010) [110,115,118].

The AVE, which explains the convergent validity of each item towards its construct, is greater than 0.5 for all items. This is good, as it indicates that each construct explains over half of the variance of its indicators (Henseler, Ringle and Sincovics 2009) [118].

Table 4 presents the results of the discriminant validity.

**Table 4.** Discriminant validity.

| | Use of Ecological Charcoal | Ecological Sensitivity | Economical Sensitivity | Health and Security | Quality Cleanliness |
|---|---|---|---|---|---|
| Use of Ecological Charcoal | 0.942 | | | | |
| Ecological Sensitivity | 0.258 | 0.781 | | | |
| Economical Sensitivity | 0.281 | 0.355 | 0.781 | | |
| Health and Security | 0.291 | 0.450 | 0.355 | 0.857 | |
| Quality and Cleanliness | 0.279 | 0.278 | 0.266 | 0.526 | 0.792 |

The discriminant validity is checked through the Fornell–Larcker criterion. It suggests that the square root of the AVEs, along with the diagonal, must be greater than all the values below and left of the AVEs located on the diagonal (Fornell–Larcker, 1981) [110]. This test does not detect any anomaly, as displayed in Table 4. Each construct shares more variance with its own block of indicators than with other latent variables represented by a different block of indicators (Henseler, Ringle and Sincovics 2009) [110]. Thus, we conclude that discriminant validity is not an issue in this research model.

*5.3. Assessment of the Structural Model*

To test the hypotheses raised in this research and assess the link between the variables, we make use of the SmartPLS 4 software. This software allows the examination of a series of dependency relationships (Bollen 2014; Ringle, Wende, and Becker 2015; Mfokeu and Chrysostome 2021) [109,119,120], which are represented graphically in Figure 3.

Additionally, the results of the estimation are presented in Table 5.

**Table 5.** Hypotheses tests.

| Hypotheses | Original Sample (O) | Sample Mean (M) | Standard Deviation (STDEV) | T Statistics (|O/STDEV|) | *p* Values | Decision |
|---|---|---|---|---|---|---|
| ECOLOGICAL SENSITIVITY → ECOLOGICAL CHARCOAL | 0.111 | 0.114 | 0.039 | 2.808 | 0.005 | Accepted |
| ECONOMIC SENSITIVITY → ECOLOGICAL CHARCOAL | 0.166 | 0.168 | 0.055 | 2.994 | 0.003 | Accepted |
| HEALTH and SECURITY → ECOLOGICAL CHARCOAL | 0.103 | 0.103 | 0.054 | 1.891 | 0.044 | Accepted |
| QUALITY and CLEALINESS → ECOLOGICAL CHARCOAL | 0.150 | 0.155 | 0.046 | 3.244 | 0.001 | Accepted |

It appears from Table 5 and Figure 3 that the coefficient associated with ecological sensitivity is positive and significantly different from zero. This means that ecological sensitivity has a positive influence on the use of ecological charcoal. Thus, our first hypothesis is validated.

Similarly, the coefficient associated with economic sensitivity is positive and significantly different from zero. The economic sensitivity also positively influences the use of ecological charcoal. Our second hypothesis is also validated.

Third, the coefficient associated with health and security is positive and significantly different from zero. The desire to preserve health and security influences the use of ecological charcoal. Our third hypothesis is also validated.

Finally, the coefficient associated with quality and cleanliness is positive and significantly different from zero. Preference for the quality of meals and the cleanliness of pots has a positive impact on the use of ecological charcoal. Our fourth hypothesis is thus validated.

In sum, the estimation of the structural model leads to the conclusion that the consumers of ecological charcoal in Cameroon are motivated by several factors, which include the desire to protect the environment (ecological sensibility); the desire to reduce the energy costs for cooking (economic sensibility); the desire to preserve health and security; and the preference for better quality meals and cleanliness of pots.

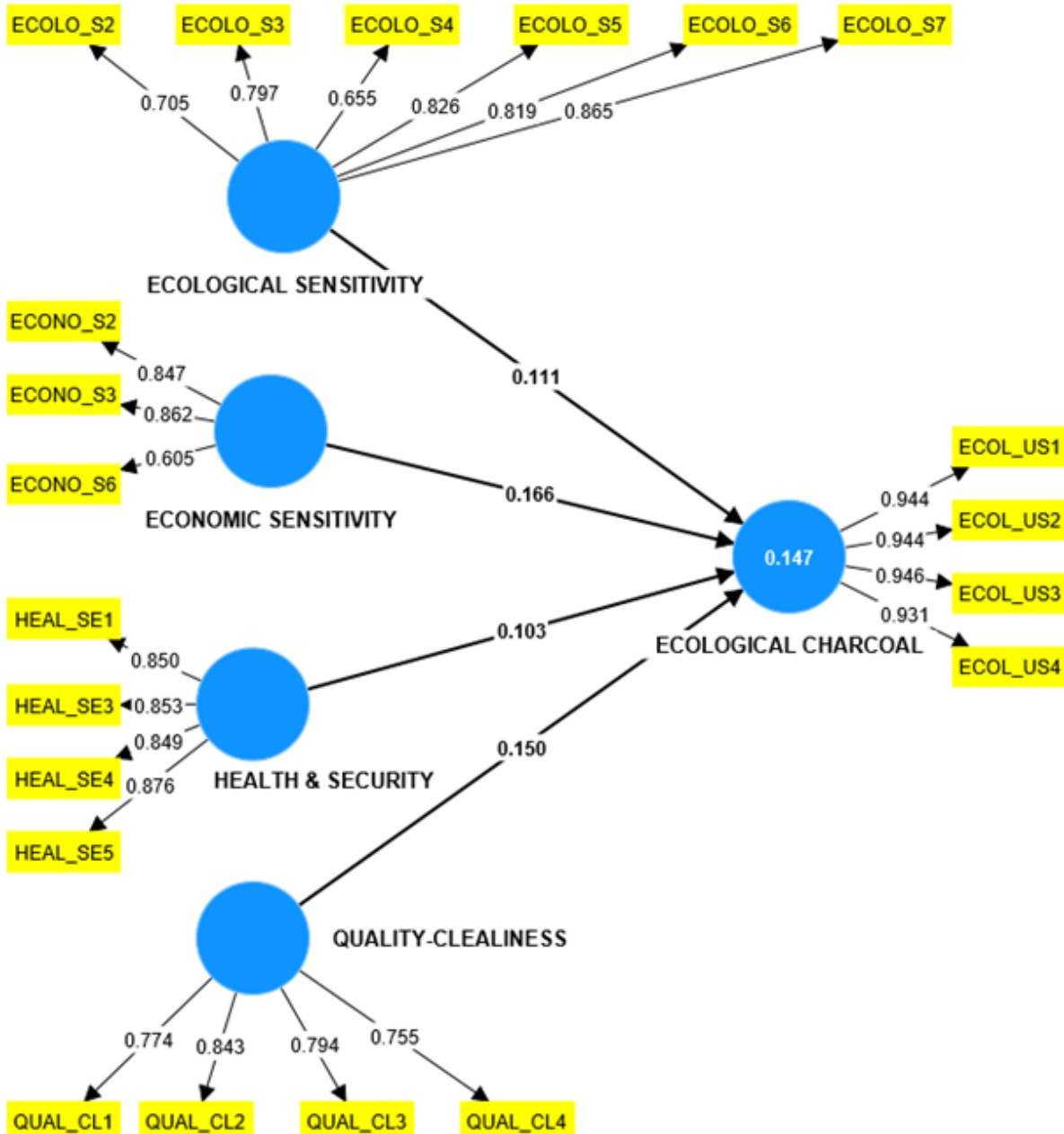

**Figure 3.** Structural equation modeling results.

*5.4. Discussions*

The main objective of this research is to analyze the motivation for the consumption of ecological charcoal in Cameroon. The results show that the main factors explaining respondents' use of ecological charcoal are the desire to protect the environment (ecological sensibility), the desire to reduce the costs of energy (economic sensibility), the need to improve health and security, and the desire to preserve the quality of meals and the cleanliness of pots. These variables are all correlated with the use of ecological charcoal.

*5.5. Economic Sensitivity*

The results show that the need to save money positively and significantly affects the probability of using ecological charcoal. In other words, compared to other cooking energy mechanisms (gas, wood, traditional charcoal, griddle . . . ), the reduction in costs associated with ecological charcoal increases the probability of its use. In addition, compared to the other energy sources mentioned, it appears that the use of ecological charcoal increases the

probability of saving money. Individuals would benefit from using ecological charcoal to save money given the strong household pressure on wood resources, which tends to cause shortages in the market (Tchatat 2017) [121]. In the energy transition, ecological charcoal occupies the first rung of the ladder, representing the cheapest energy choice. Financially constrained consumers will tend to favor this solution. As the sample comprises 36.95% of people with low income (less than CFA 50,000 monthly revenue), the choice of an energy source corresponds for this population to the satisfaction of a basic need in Maslow's hierarchy of needs. This result is in line with the work of Hosier and Dowd (1987) [50] and Reddy and Reddy (1994) [57], who show that there is a hierarchical ladder of energy sources, where each income level has a dominant type of energy that maximizes the utility of households.

### 5.6. Ecological Sensitivity

The results also show that the need to protect the environment positively and significantly affect the probability of using ecological charcoal. This result is refreshing. In the sample of consumers surveyed, there is a feeling of wanting to protect the environment. It is strong enough to motivate consumers to opt to use ecological charcoal as a means of cooking. The mass production of ecological charcoal provides a solution to several problems that arise not only in Cameroon, but in Africa in general. Waste management is a real problem in Cameroon. In various cities of the country, it is far from efficient. The arteries of these cities are full of rubbish, and the municipalities are unable to solve the problem. In most cases, these municipalities face the limitation of financial resources. As raw materials from waste can be used in the production of ecological coal, a good part of the waste would thus find a natural outlet. A monthly production of 15 tons of ecological coal requires 240 tons of household waste. This would reduce the landfilling of waste, which creates its own share of pollution. The intensive use of ecological coal would make it possible to fight against deforestation. For the far north region alone, Kodji et al. (2021) [13] concluded that: " ... a substitutional energy at low cost and accessible would be ecological charcoal. The Region has an estimate annual production capacity at 2.65 million tons in average of ecological charcoal, the equivalent of about 60,000TJ. Such production could cover the annual needs of 10 million inhabitants right up to 2050 in the Region." Extrapolating this projection over the whole country indicates that a lot of trees could be saved in the forest.

Concerns about the environment are not usually one of the primary needs for individuals living in Cameroon. Usually, ecological products are more expensive than "ordinary" ones, which means consumers pay an environmental premium to use them. The advantage of ecological charcoal is that it is a low-cost, environmentally friendly source of energy for domestic energy cooking.

### 5.7. Need to Improve Health and Security

The results show that the need to improve health and security positively and significantly affects the probability of using ecological charcoal. The inefficient burning of woody biomass is responsible for acute respiratory infections causing over 600,000 premature deaths per year in Africa (WHO 2021) [2]. As indicated above, health problems related to exposure to indoor air pollution can lead to difficulties in carrying out activities such as education, domestic tasks and labor market participation. This in turn can generate economic losses, including lost wages and increased medical costs (Bakehe 2021) [105]. Beyond the ecological act, consumers of ecological charcoal want to reduce this air pollution significantly. At the same time, these consumers reduce the risk of burns and fire, which means increased security.

### 5.8. Need to Preserve the Quality of Meals and Ensure the Cleanliness of Pots

The results of the study show that the effects of quality and cleanliness on the use of ecological charcoal are positive and significant. This means that, additionnaly to the

cleanliness of pots that it involves, consumers tend to believe that cooking with ecological charcoal results in better-quality meals.

The results of this paper are in line with those of other researchers looking for solutions that can help the planet achieve the United Nations Sustainable Development Goal (SDG) number 7. Perez et al. (2020) [122] have proposed an improved stove based on gasification, as well as briquettes obtained by the carbonization and densification of agricultural wastes. They have documented a fuel saving of 61% and a decrease of 20% in the time used for cooking when the improved stove is used rather than traditional charcoal. Similar fuel savings are obtained when using the briquettes, with an 18% reduction in the time used for cooking. These translate into savings in the costs of energy, and an emission reduction in $CO_2$.

The results of this paper have managerial and policy implications.

Regarding managerial implications, first, there is a huge market for ecological charcoal in Cameroon. Currently, the product is not well known by consumers, and it is not widely available. Efforts should be made to increase the production and distribution of ecological charcoal within the country. Second, consumers attach importance to the selling price and choose ecological charcoal as a means to keep their cooking costs low. Thus, firms offering the product should endeavor to keep the price affordable for low-income consumers. Third, the selling price does not need to be the same for all customers. A market segmentation may be helpful for supplying firms. Environmentally convinced consumers are usually ready to pay more for ecologically friendly products. They will probably be ready to pay more for ecological charcoal. Marketing strategies targeting this market segment can be considered. The market segmentation invites firm managers to take into account the differences between consumers and to better target their communication. In concrete terms, communication about ecological charcoal should emphasize the message about the economic aspect of the product, while also conveying the ecological aspect. Such a strategy would allow different categories of buyers to identify with green charcoal. For example, the economically sensitive consumer will be satisfied with the price of the product, and will buy the lower-price packaging. The environmentally sensitive consumer may be interested in the ecological aspect of the product, and then concede to pay a higher price. Fourth, public announcements should always put forward the environmental value of ecological charcoal.

As for policy implications, the results of this study call for the attention of public decision-making authorities in Cameroon and other developing countries. Protecting the environment is a duty for them, and ecological charcoal offers a means to contribute to reaching this goal. Ecological charcoal is not yet well known in the country. The government and municipalities will have to be forward-minded. This can be done through several actions: First, the government should contribute to the popularization of this product, by subsidizing the entrepreneurs who want to invest in the production of ecological charcoal. This will increase the offers on the market, and make the product available in large quantities to consumers. Yet, ecological charcoal is not widely available within the country. Second, the government and municipalities should promote the use of ecological charcoal, through advertisement campaigns within the country. Economical charcoal is not yet well known by all consumers. The high consumption of ecological charcoal will help cities and municipalities alleviate their waste management problems. Ecological charcoal is well appreciated by the consumers who have had the opportunity to try it, but it remains little known and not easily available to the vast majority. Public authorities should support its industrial production and promotion. This will have several positive consequences, the best known of which offering the beginning of a solution to the glaring problem of waste collection by municipalities and the protection of the environment.

The conceptual model formulated and presented in Figure 2 forms a strong theoretical basis, which integrates four important dimensions of ecological charcoal: the economic, the ecologic, the heath and security and the quality and cleanliness dimensions. The four hypotheses derived from this framework are all validated, highlighting the potential of ecological charcoal to play an important role in energy transition, specifically in developing

countries. As such, the study acts as a useful resource for practitioners. Access to clean and affordable energy is a basic human need and is emphasized in the United Nations Sustainable Development Goal (SDG) 7 ''Ensure access to affordable, reliable, sustainable and modern energy for all'' (United Nations, 2022) [123]. Ecological charcoal can help achieve this goal.

## 6. Conclusions

The environmental record of the world is not rosy. The planet sends us alarming signals on a daily basis: global warming, desertification, melting glaciers, hurricanes, torrential rains, landslides, natural disasters, etc. As such, protecting the environment is a global concern, and any positive action around the world counts.

This paper has analyzed consumers' motivations to use ecological charcoal (an environment-friendly product) in Cameroon. To reach this end, data from a sample of 525 consumers and the PLS-SEM methodology were used. The estimation of the structural model led to the conclusion that in Cameroon, the consumption of ecological charcoal is motivated by several factors. These include the desire to protect the environment (ecological sensibility), the desire to reduce the energy costs of cooking (economic sensibility), the need to improve health and security, and the desire to enhance the quality of meals and preserve the cleanliness of pots.

Although the supply of ecological charcoal is still limited across the country, it is refreshing to know that in addition to its economic character, consumers choose it for its ecological virtues. Its massive consumption will contribute to a reduction in household waste management problems in the cities and municipalities, while preserving the environment. Economical charcoal is a means that can help achieve the United Nations' SDG 7.

Even though this research makes novel contributions to the literature, it has some limitations. Only two cities were involved in this study: Yaoundé and Douala. Thus, the study may not accurately reflect the behavior of the entire population of Cameroon. With less educated and less wealthy people, the population living in smaller cities may behave differently.

It should be mentioned that the results of this research may be influenced by the socio-demographic characteristics of the respondents, and may require some verification before generalization. Thus, a study that takes into account several other cities with a larger number of respondents could better explain the motivations for the consumption of ecological charcoal, and highlight the differences between the segments of this market. Additionally, as a continuation of this work, it would be interesting to extend this issue to other countries with the same cultural environment.

**Author Contributions:** Conceptualization, A.M.M. and O.E.M.N.; methodology, J.-P.G. and A.M.M.; software, A.M.M.; validation, A.M.M., E.V.C., J.-P.G. and O.E.M.N.; formal analysis, A.M.M. and J.-P.G.; data curation, A.M.M. and O.E.M.N.; writing—original draft preparation, A.M.M., E.V.C. and O.E.M.N.; writing—review and editing, A.M.M., E.V.C., J.-P.G. and O.E.M.N.; supervision, J.-P.G.; project administration, A.M.M. All authors have read and agreed to the published version of the manuscript.

**Funding:** This research received no external funding.

**Institutional Review Board Statement:** Ethical review and approval were waived for this study, due to the fact that informed consent was obtained from all subjects involved in the study.

**Informed Consent Statement:** Informed consent was obtained from all subjects involved in the study.

**Data Availability Statement:** The datasets used and/or analyzed during the present study are available from the corresponding author upon reasonable request.

**Conflicts of Interest:** The authors declare no conflict of interest.

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
