# Peer review of "Consumer Motivation behind the Use of Ecological Charcoal in Cameroon"

_sustainability, doi:10.3390/su15031749_

Round 1

Reviewer 1 Report

The topic is really interesting and the methodology clearly explained. In section "3. Literature review and hypotheses development", some similar experience could be reported or on the topics of green coal as a resource from food or agriculture and the circular economy as an approach to ecology and energy transition.

Author Response

Dear Reviewer.

Reviewer 2 Report

Kindly follow the highlights and apply accordingly.

Reviewer 3 Report

How does this study's underlying theory explain the inclusion of multiple determining factors for the usage of ecological charcoal in Cameron? If not, the addition of variables appears somewhat random. It requires additional and improved theory for selecting variables and their relationships. There are insufficient arguments as to why these variables should be integrated when examining the determinants of ecological charcoal use among Cameroonians. The authors cited the Theory of Planned Behavior in general. Still, they made no connection between it and the construction of their conceptual framework, nor did they connect the variables to any of the original TPB dimensions.

In addition, the authors must provide sufficient justification for the inclusion of the moderator and its contribution to the conceptual model.

The development of hypotheses is an essential step in quantitative research. The authors didn’t deduce the moderating hypotheses as illustrated in Fig.1. for instance: 

H5a: The positive relationship between ecological sensitivity and the use of ecological charcoal is stronger/weaker when the income level is high/low. 

The same holds for H5b in terms of economic sensitivity. For additional reading on how to build the moderation hypothesis, see Dawson (2014).

How did the researcher decide on the sample size? This was not discussed.

Taste, quality, and cleanliness should be viewed as formative latent variables and not reflecting variables in the evaluation of the measurement model; hence, the assessment should be different. If not, explain why the variable is reflective and not formative when it consists of three sub-dimensions and excluding a statement from any of these sub-dimensions renders the variable unable to be measured and reflected upon as a whole.

How did the authors divide the income into high-income and low-income groups to evaluate the interaction effect between hypotheses? In accordance with the usual procedure, moderators should be divided into two groups and labeled 0 and 1 prior to SmartPLS execution. The current result does not indicate any interaction, and it is unclear whether the moderator has strengthened or weakened the association between ecological sensitivity, economic sensitivity, and the use of ecological charcoal.

Discussion can be improved. Weak at the moment, just reporting the result and comparing it to previous literature! The suggestion is also brief. It would be helpful if the authors could provide a more concrete discussion of the implications of these results. The current discussion is somewhat vague. The author must provide more information and detail, particularly regarding the model's theoretical contributions and how it acts as a useful resource for practitioners.

Round 2

Reviewer 3 Report

Accept in current form.